# How Do Older Patients with End-Stage Osteoarthritis of the Hip Eat Prior to Hip Replacement? A Preliminary Snapshot That Highlights a Poor Diet

**DOI:** 10.3390/nu15234868

**Published:** 2023-11-22

**Authors:** Matteo Briguglio, Paolo Sirtori, Laura Mangiavini, Sara Buzzi, Claudio Cordani, Maria Francesca Zerni, Thomas W. Wainwright, Nicola Ursino, Giuseppe M. Peretti, Giuseppe Banfi

**Affiliations:** 1Laboratory of Nutritional Sciences, IRCCS Orthopedic Institute Galeazzi, Via Riccardo Galeazzi 4, 20161 Milan, Italy; 2E.U.O.R.R. Unit, IRCCS Orthopedic Institute Galeazzi, Via Riccardo Galeazzi 4, 20161 Milan, Italygiuseppe.peretti@unimi.it (G.M.P.); 3Department of Biomedical Sciences for Health, University “La Statale” of Milan, 20133 Milan, Italy; 4Department of Biomedical, Surgical, and Dental Sciences, University “La Statale” of Milan, 20141 Milan, Italy; claudio.cordani@unimi.it; 5IRCCS Orthopedic Institute Galeazzi, Via Riccardo Galeazzi 4, 20161 Milan, Italy; 6Orthopaedic Research Institute, Bournemouth University, Bournemouth BH8 8FT, UK; 7Physiotherapy Department, University Hospitals Dorset NHS Foundation Trust, Bournemouth BH7 7DW, UK; 8C.A.S.C.O. Unit, IRCCS Orthopedic Institute Galeazzi, Via Riccardo Galeazzi 4, 20161 Milan, Italy; 9Faculty of Medicine and Surgery, Vita-Salute San Raffaele University, Via Olgettina 58, 20132 Milan, Italy

**Keywords:** prevalence study, orthopaedic surgery, total hip arthroplasties, pre-operative care, eating behaviour, dietary habits, healthy eating index, protein calorie malnutrition, macronutrient intake, nutritional status

## Abstract

Diet quantity and quality in older adults is critical for the proper functioning of the musculoskeletal system. In view of hip surgery, old patients should consume 1.2–1.5 g of proteins and 27–30 kcal per kilo of body weight daily, and adhere to healthy eating habits. In this analytical study, we studied diet quantity and quality in relation to the clinical chemistry and functional status of 57 older adults undergoing elective hip replacement. Nine in ten patients did not meet suggested protein and energy intakes and only one in ten patients exhibited high adherence to the Mediterranean diet. Legume consumption adjusted for sex, age, body mass index, and health status successfully forecasted haemoglobin levels (*p* < 0.05), and patients regularly consuming olive oil reported minor hip disability compared to those using it less frequently (*p* < 0.05). Patients who reported daily ingestion of <1 serving of meat versus those consuming >1.5 servings had greater cumulative comorbidity (*p* < 0.05), with meat consumption independently predicting walking ability, mobility, and balance in the fully adjusted model (*p* < 0.01). In conclusion, our patients seem to eat poorly. There is room for improvement in pre-operative pathways to make older adults eat better, but there is a need to plan an interventional study to fully understand the cause–effect of a dietary pattern or specific food in enhancing recovery after surgery.

## 1. Introduction

Poor diet refers to insufficient quantity and low quality. Food quantity mainly refers to the concept of adequate servings for body weight, while quality echoes the variety of food choices [1]. Low dietary scores were shown to be present in older people with reduced gait speed [2] or osteoarthritis [3,4], being suggested to increase the odds of fragility fracture [5]. Health initiatives have been emerging in recent years involving community-dwelling people to promote health through multimodal approaches [6], including nutrition advice. Weight control, Mediterranean diet, and lipid modification towards olive oil and oily fish are some nutritional factors that are currently believed to reduce the progression of osteoarthritis [7,8]. While it is plausible to think that healthy eating education could be sufficient at the community level, a more specialist approach may be needed for those patients with illness or undergoing an operation [9]. Tailored advice on what to eat (diet therapy, nutritional supplements) is important to increase suitability for surgery, potentially enhance recovery, and return independence to patients, especially if old and malnourished [10]. This is one of the three pillars of prehabilitation (exercise, nutrition, psychology) that is increasingly becoming utilised in pre-operative interventions [11]. This shall be decisive for the older population undergoing hip surgery, where adequate intake of proteins (1.2–1.5 g pro kilo of body weight) and energy (27–30 kcal pro kilo of body weight) could be critical to reduce the risk of complications and ensure proper recovery [1]. For this specialist initiative to be effective, there is a need to know how older individuals eat in view of hip replacement. Additionally, it is not uncommon for older patients to ask health professionals what to eat to enhance recovery [1], implying that patients are not sure what diet to follow. If it were to be demonstrated that patients who follow a healthy diet before surgery are the healthiest, there would be a basis for integrating clinical practice with nutrition advice. This brief report presents the eating habits of old patients undergoing elective hip replacement to explore the appropriateness of both protein–energy intakes and adherence to the Mediterranean dietary pattern and to examine the existence of an association between diet quantity, diet quality, and health status.

## 2. Materials and Methods

### 2.1. Study Design and Eligibility Criteria

This is a preliminary investigation from a prospective analytical study that involved 60 older patients undergoing elective surgery of the hip performed between 2019 and 2022 at IRCCS Orthopedic Institute Galeazzi of Milan, Italy (NCT03981354). The study was planned to explore the role of nutritional status indicators, such as diet, in determining the strength of constitution, suitability for surgery, and recovery in the older population undergoing major orthopaedic surgery. Screening involved candidates for elective hip replacement during the pre-admission visit at our hospital. Older adults (between 60 and 85 years old), males or females, and with the indication to undergo primary total hip replacement were eligible to participate. We excluded those with heart or kidney failure, cancer, neuropsychiatric disorders on medications for major conditions, and inability to adhere to the protocol or to carry out the evaluations. After recruitment, subjects undergo assessments for laboratory parameters, anthropometry, eating behaviours, and physical performance. Processing of the complete study data is still in progress. For the purpose of this preliminary report, we present the results concerning pre-operative dietary indexes, blood analytes, and functional status.

### 2.2. Dietary Assessment and Laboratory Parameters

As a measure of diet quantity, we recalled the food and beverages consumed in the last 24 h (24HR) to obtain daily proteins·kg^−1^ and kilocalories·kg^−1^. Diet quality was assessed with the Medi-Lite index [12], which records how many times a person reports daily or weekly consuming nine food groups, specifically fruit, vegetables, legumes, cereals, fish, meat and meat products, dairy products, alcohol, and olive oil. The score assigned to the answers on fruit, vegetables, legumes, cereals, fish, and olive oil was higher the more frequent the consumption. Conversely, the lower the meat and cheese consumption, the higher the score. The highest-rated answer on alcohol consumption was 1–2 units per day. The Medi-Lite scores from 0 to 18, with higher values being representative of greater adherence to the Mediterranean diet pattern. The following blood analytes known to be associated with an individual’s nutritional status were retrieved: red blood cells (RBCs), haematocrit, haemoglobin, mean corpuscular volume (MCV), mean corpuscular haemoglobin (MCH), mean corpuscular haemoglobin concentration (MCHC), iron, transferrin, ferritin, white blood cells (WBCs), lymphocyte count, lymphocyte percentage, transthyretin, albumin, total proteins, and C-reactive protein (CRP).

### 2.3. Functional Status

For measuring the ability of patients to daily care for themselves, we used the 10-item Barthel index (BI) that assesses the level of independence to eat, bath, groom, dress, reach and use the bathroom (bowel and bladder control, toileting), move from bed to chair and back, ambulate on level surfaces, and climb the stairs. The BI scores from 0 to 100, with the highest scores indicating independence. To objectively assess walking ability, mobility, and balance, we used the timed up and go test (TUG) that measures the seconds a patient takes to rise from a chair, walk 3 metres, turn, and walk back to sit on the same chair. As a measure of outcome reported by the patient, we recorded how well patients reported being able to perform five different daily activities (descending stairs, getting in/out of bath or shower, sitting, running, twisting/pivoting on the loaded leg) with the short form of the condition-specific tool hip disability and osteoarthritis outcome score-physical function (HOOS-PS) [13]. The 5-item HOOS-PS spans from 0 to 100, with the lowest score indicating absence of impairment. Patients were also asked to respond to the generic 12 items of the short-form health survey from which the physical health composite summary (SF-12 PCS) was obtained. Higher scores of the SF-12 PCS indicate better physical functioning from the patient’s perspective.

### 2.4. Statistics

As a measure of cumulative morbidity, we calculated the Charlson comorbidity index (CCI) [14]. To estimate protein and energy intakes from the 24HR, we used the on-line food composition database for epidemiological studies in Italy (BDA; bda-ieo.it accessed between February and July 2023). Cohort descriptors were planned to be reported as frequencies when categorical or as mean and standard deviation when continuous. Inferential analyses were planned to investigate the association between diet quantity (daily proteins·kg^−1^ and kilocalories·kg^−1^) and quality (Med-Lite index) with laboratory parameters and functional scores. The rank-based nonparametric Mann–Whitney U test was used to determine if there were sexual differences in the descriptors (asymptotic *p*-value reported). The Spearman’s rank-order correlation tested the existence, strength, and direction of the potential association between dietary scores and the other descriptors. The following continuous variables were ranked into quartiles: protein·kg^−1^ (Q1 = 0.56; Q2 = 0.71; Q3 = 0.93), kilocalories·kg^−1^ (Q1 = 15.47; Q2 = 18.96; Q3 = 23.83), and Med-Lite (Q1 = 9; Q2 = 10; Q3 = 12). We ran the rank-based nonparametric Kruskal–Wallis H test to determine if there were differences in the distributions of the descriptors between the following four groups: ≤Q1, >Q1 and <Q2, ≥Q2 and <Q3, and ≥Q3. The Mann–Whitney U test was applied to confirm differences across dietary quartiles and the scores of the nine food groups from the Medi-Lite. Prediction models assumed that diet is the exposure (independent variable) and the clinical status is the outcome (dependent variable). Hierarchical multiple regression was run to verify the prediction potential of sex, age, BMI, CCI (block 1), and dietary indexes (block 2) on laboratory or functional descriptors. Dietary variables were therefore planned to be added in Model 2 to observe if the change in explanatory power was statistically significant. Analyses were performed with the software IBM SPSS Statistics v.22 and all tests showed a statistically significant difference if *p* < 0.05.

## 3. Results

### 3.1. Descriptives, Associations, and Differences in Cohort Descriptors

A total of 60 patients were recruited at pre-admission visits, with 3 being excluded because of surgery deferral. The cohort (Table 1) was composed of 47.37% females, showing sexual differences for body weight (*p =* 0.0001), protein·kg^−1^ (*p =* 0.026), kcalories·kg^−1^ (*p =* 0.026, U = 188, z = −2.224), and RBCs (*p =* 0.016). There was a correlation between proteins·kg^−1^ and haematocrit (r_s_ = −0.299, *p =* 0.037) and CRP (r_s_ = −0.405, *p =* 0.004). Kilocalories·kg^−1^ significantly correlated with CRP (r_s_ = −0.481, *p* < 0.001). Medi-Lite score significantly associated with the HOOS-PS (r_s_ = 0.410, *p =* 0.013). For the food groups, we found the following correlations: fruit and age (r_s_ = 0.285, *p =* 0.042); legumes and haemoglobin (r_s_ = 0.285, *p =* 0.042), legumes and CRP (r_s_ = −0.293, *p =* 0.039); cereals and total proteins (r_s_ = −0.385, *p =* 0.006); meat and TUG (r_s_ = 0.331, *p =* 0.026); meat and HOOS-PS (r_s_ = 0.455, *p =* 0.005); dairy and lymphocyte percentage (r_s_ = −0.277, *p =* 0.049); dairy and total proteins (r_s_ = 0.476, *p <* 0.001); olive oil and TUG (r_s_ = −0.399, *p =* 0.007); and olive oil and HOOS-PS (r_s_ = −0.360, *p =* 0.031).

Distribution of CRP was different between quartiles of proteins·kg^−1^ (*p* = 0.033), kilocalories·kg^−1^ (*p* = 0.021), and Medi-Lite (*p* = 0.009). Also, distribution of HOOS-PS was different between Medi-Lite quartiles (*p* = 0.023). Concerning the food groups, we found different distributions of haematocrit for vegetables (*p* = 0.035), TUG for vegetables (*p* = 0.012), total proteins for cereals (*p* = 0.026), haemoglobin for meat (*p* = 0.047), HOOS-PS for meat (*p* = 0.013), total proteins for dairy (*p* = 0.003), TUG and olive oil (*p* = 0.028), and HOOS-PS and olive oil (*p* = 0.033). CRP was significantly different between Q1 and Q4 (*p =* 0.011) and Q2 and Q4 (*p =* 0.016) of proteins·kg^−1^, between Q1 and Q4 (*p =* 0.003) of kilocalories·kg^−1^, between Q1 and Q2 (*p =* 0.017), Q3 and Q4 (*p =* 0.017), and Q1 and Q4 (*p =* 0.002) of Medi-Lite, and <1 and >2 weekly servings of legumes (*p =* 0.038). HOOS-PS was significantly different between Q2 and Q3 (*p =* 0.012), Q1 and Q3 (*p =* 0.013) of Medi-Lite. Between 1–2.5 and > 2.5 daily servings of vegetables, we found a statistically significant difference in haematocrit (*p =* 0.007), haemoglobin (*p =* 0.04), and TUG (*p =* 0.003). There was also a significant difference between 1–1.5 and >1.5 daily serving of cereals for circulating total proteins (*p =* 0.016). WBCs were different between <1 and 1–1.5 weekly serving of fish (*p =* 0.044) and transferrin was different between 1–2.5 and >2.5 weekly servings (*p =* 0.045). CCI was different between <1 and >1.5 daily servings of meat (*p =* 0.038), similar to TUG (*p =* 0.030). Between <1 and 1–1.5 daily servings of meat, there was a difference in haemoglobin (*p =* 0.018) and MCHC (*p =* 0.039). HOOS-PS was different between >1.5 and <1 (*p =* 0.004), 1–1.5, and <1 daily serving of meat (*p =* 0.042). Concerning dairy, we found a difference in total proteins between >1.5 and 1–1.5 (*p =* 0.008) and >1.5 and <1 daily servings (*p =* 0.003), and in iron (*p =* 0.027). There was also a difference in lymphocyte percentage between >1.5 and <1 daily servings of dairy (*p =* 0.029). Between regular and frequent use of olive oil, there was a significant difference in TUG (*p =* 0.02) and HOOS-PS (*p =* 0.033).

### 3.2. Predictions of Clinical Conditions from Dietary Scores

The significant results of sequential multiple regressions are reported in Table 2. We did not observe a statistically significant increase in explanatory power or in the fully adjusted model using proteins·kg^−1^, kcalories·kg^−1^, Medi-Lite index, fruit, vegetables, fish, or alcohol consumption. However, the addition of legumes to the prediction of haemoglobin led to a significant increase in R^2^ of 0.072 (*p* = 0.043). The full model with sex, age, BMI, CCI, and legumes predicted haemoglobin levels (*p* = 0.017, adjusted R^2^ = 0.175). The fully adjusted model with cereals predicted total proteins (*p* = 0.024, adjusted R^2^ = 0.162), with the addition of cereals statistically increasing R^2^ to 0.179 (*p* = 0.002). The full model of sex, age, BMI, CCI, and meat predicted the TUG (*p* = 0.017, adjusted R^2^ = 0.199), increasing R^2^ of 0.143 (*p* = 0.008). Meat scores were also revealed to contribute to the statistically significant increase in R^2^ of 0.203 (*p* = 0.002) for what concerned the model of HOOS-PS, with the fully adjusted model statistically predicting the HOOS-PS (*p* = 0.001, adjusted R^2^ = 0.397. Concerning the dairy variable, its addition to the prediction of total proteins led to a statistically significant increase in R^2^ of 0.179 (*p* = 0.002). The fully adjusted model was also statistically significant (*p* = 0.025, adjusted R^2^ = 0.161).

The fully adjusted model with olive oil consumption to predict TUG was statistically significant (*p* = 0.019, adjusted R^2^ = 0.192). The addition of olive oil to the prediction of TUG led to a statistically significant increase in R^2^ of 0.137 (*p* = 0.009). Similarly, olive oil consumption statistically significantly increased R^2^ to 0.091 (*p* = 0.046) of the full model that predicted HOOS-PS (*p* = 0.012, adjusted R^2^ = 0.266). No statistically significant associations were found with RBCs, MCV, MCH, ferritin, transthyretin, albumin, Barthel index, SF-12 PCS, fruit, or alcohol consumption.

## 4. Discussion

We investigated the eating habits reported by 57 old patients before primary hip replacement in order to (1) quantify whether protein and energy intakes were sufficient, (2) verify how much their dietary lifestyle adhered to the Mediterranean diet, and (3) investigate whether there was a relationship between diet and clinical condition.

According to the conventional definition of BMI categories, our cohort included 20 patients with normal weight, 24 overweight, 9 obese I, 3 obese II, and 1 obese III. When considering the more recent BMI categorisation for older adults (<22.0 kg·m^−2^ = underweight; 22.0–28.4 kg·m^−2^ = normal; 28.5–33.4 kg·m^−2^ = overweight; 33.5–38.4 kg·m^−2^ = obesity I; 38.5–43.4 kg·m^−2^ = obesity II; ≥43.5 kg·m^−2^ = obesity III) [15], we would count 8 as underweight, 31 normal weight, 10 overweight, 7 obese I, and 1 obese II. Since we found that over 90% of patients of this cohort did not meet the protein and energy intakes required for surgical patients, the latter BMI classification is more consistent with our findings, showing 14% undernourished individuals. Adequate energy and protein intake in older people is known to be critical for the function of the immune system, the counteraction of the physiological decline of the musculoskeletal system, and balancing of intestinal malabsorption with reduced biodigestibility [16]. We observed sexual differences in both energy (18.01 kcal in males and 21.46 kcal in females) and protein (0.69 g in males and 0.84 g in females) intake per kilogram of body weight, but this could be ascribed to significant body weight differences. Mean weight in females was 67.64 kg versus 83.46 kg in males, with the recall of food consumed in the last 24 h estimating similar intakes: females reported ingesting approximately 1396 kcalories and 55 g of proteins versus 1489 kcalories and 58 g of proteins in males. Concerning diet quality, the Mediterranean diet is currently one of the dietary patterns suggested to manage osteoarthritis, mainly due to its anti-inflammatory and antioxidant properties [7]. It mainly includes fruit, vegetables, whole grains, fish, and legumes, limiting the consumption of meat and dairy products. Regular use of olive oil is advised, and moderate alcohol consumption is allowed. In our cohort, no sexual differences in adherence to this dietary model were found. Overall, only 10% exhibited high adherence to the Mediterranean diet (Medi-Lite score ≥ 13). All other patients reported a medium adherence (Medi-Lite score ≥ 6 and <12).

In our cohort, CRP levels were significantly different (*p* < 0.01) between patients with the highest (Q4, score 13–15 of Medi-Lite) and lowest (Q1, score 6–9 of Medi-Lite) adherence to the Mediterranean diet being 0.08 mg·L^−1^ and 0.38 mg·L^−1^, respectively. We also found statistically significant relationships between CRP and 24HR protein and energy intakes. Patients reporting eating the highest amount of protein (Q4, 0.94–1.57 g pro kilo) had a mean CRP of 0.18 mg·L^−1^ that was significantly lower (*p* < 0.05) than 0.62 mg·L^−1^ in those eating the lowest quantity (Q1, 0.31–0.56 g pro kilo). Similarly, patients with the lowest intakes of energy (Q1, 9.79–15.47 kcal pro kilo) had a CRP value of 0.66 mg·L^−1^, significantly higher (*p* < 0.01) than 0.18 mg·L^−1^ in those consuming the greatest amount (Q4, 23.99–36−47 kcal pro kilo). Patients eating more than 2.5 servings of vegetables a day showed significantly lower levels of haematocrit (40.53%) and haemoglobin (13.81 g·dL^−1^) compared to those consuming 1–2.5 servings (haematocrit: 43.52%; haemoglobin: 14.47 g·dL^−1^). However, this could be attributed to sexual differences in RBCs, since females, who are known to exhibit significantly lower levels of erythrocytes, were the majority in the group consuming more vegetables. Both cereals and dairy consumption were valuable independent variables in making predictions of circulating total proteins. When reducing the consumption of cereals from >1.5 to 1–1.5 or <1 daily serving, or of dairy from >1.5 to 1–1.5 or <1 daily serving, there was an increase in total proteins of about 0.36 g·dL^−1^ and 0.28 g·dL^−1^, respectively. Interestingly, the lower the dairy consumption, the lower the percentage of lymphocyte: 24.39% when <1 serving, 26.52% when 1–1.5 serving, and 30.36% when >1.5 daily serving. The potential modulatory effect of dairy products on the immune system, particularly on cellular immune function, has been known for years [17,18], but the precise biological mechanisms still remain elusive to date. When increasing the consumption of legumes from <1 to 1–1.5 or >2 weekly servings, we found an increase in haemoglobin of about 0.5 g·dL^−1^. The successful forecast of haemoglobin from the adjusted score of legumes demonstrates that this often-neglected food group is a valuable source of proteins. Beans, lentils, soy, chickpeas, and peas can be consumed after soaking and boiling if dried or after cooking if fresh or canned. Despite containing absorption disruptors (e.g., negative effectors on mineral bioavailability) and providing an unbalanced amino acid profile, when consumed with carbohydrates, legumes are considered a valuable protein-containing, nutrient-dense, and sustainable food [19].

Contrary to legumes, meat contains high-biological-value proteins similarly to fish, dairy, and eggs. Patients in our cohort that reported daily ingestion of <1 serving of meat versus those consuming >1.5 servings of meat had greater cumulative comorbidity (CCI, 3.46 versus 2.70, *p* = 0.038), worse walking ability, mobility, and balance (TUG, 13.27 versus 10.04 s, *p* = 0.03), and higher hip disability (HOOS-PS, 54.09 versus 32.19, *p* = 0.004). Eating meat and derivatives, even if not in line with the indications of Mediterranean healthy eating habits, might somehow accompany better clinical conditions, greater performance, and less disability. Patients who reported consuming <1 daily serving of meat took nearly 4 s longer on the TUG (*p* < 0.01) and had almost 23 points in the HOOS-PS (*p* < 0.01) compared to patients who ate >1.5 servings per day. Similarly, TUG and HOOS-PS were also successfully forecasted by the adjusted model with olive oil consumption. A reduction of −3 s in the TUG test and of −14 points in the HOOS-PS resulted in switching from frequent consumption to regular use of olive oil.

In Figure 1, a summary slide of the study is reported. We are not the first researchers to have studied the eating habits of patients with osteoarthritis, although, to the authors’ knowledge, this is one of the few reports on old patients with end-stage disease. Compared to a previous observational study involving American older adults undergoing total joint arthroplasty [20], we observed lower intakes of proteins and energy in the 24HR and a smaller percentage of obese patients. Other authors explored the relationship between diet and health among late-middle-aged Korean women suffering from osteoarthritis, showing worse clinical conditions in those who reported consuming low daily energy and proteins. Diet quality also independently predicted knee pain [3]. In Canadian adults with osteoarthritis aged 45–85 years, greater intakes of high-fibre cereal were associated with higher mobility score (pooled index from walking speed, one-leg strength, chair rise test, and TUG) [4]. Similarly, adhering to the Mediterranean diet and consuming a quantity of proteins higher than the recommended dietary allowance would seem to have a positive role in muscle function and physical performance, respectively, among older adults [21,22]. Although our findings from Medi-Lite total score and protein–energy intakes did not show predictive potential in the hierarchical multiple regression models, the unadjusted associations between CRP and diet quantity and quality suggest a relationship between how much and what patients reported eating and their levels of chronic inflammation. In this regard, our findings are in line with previous research showing inverse associations between CRP and intakes of either dietary flavonoids or proteins [23,24]. Divergent evidence on meat consumption and physical performance in older populations exists, including experimental trials that found no gains [25], positive results [26], or cross-sectional studies describing a negative relationship [27]. In our cohort, the association between greater consumption of meat, reduced cumulative comorbidity, and greater physical performance is in line with the suggestion that old people should eat meat a few times a week rather than never [28]. Similarly, the inverse association between olive oil consumption and hip disability is not a new finding, since the beneficial effects of olive oil and its bioactive compounds have been shown to preserve the articular cartilage and slow the progression of osteoarthritis in several animal and human studies [29,30].

### Limitations

This brief investigation has limitations. First, it is a small-sample cross-sectional study whose nature makes it unfeasible to define with certainty the direction of significant associations and to reach a valid conclusion. Second, our report is primarily based on subjective data reported by old patients, who are known to frequently under-report food intakes. Third, 24HR has intrinsic inaccuracies [31], and for this research it collected foods and beverages of the day before admission to hospital, when patients may have changed eating habits in preparation for surgery. Fourth, the Medi-Lite index neglects egg consumption, possibly removing from the equation an important animal source of high-quality proteins. Last, food timing is a matter of interest when dealing with older persons and it was not addressed in this report. Given the recommendation to distribute nutrients equally throughout the day, especially proteins [32], it is possible that smaller but sufficient amounts of proteins with each meal would bring more benefits than consuming a large daily serving.

## 5. Conclusions

This brief research report highlights modifiable deficiencies in diet amongst an older population undergoing primary hip replacement. These were linked to greater cumulative comorbidity, chronic inflammation, hip disability, and lowered pre-operative physical performance.

## 6. Future Perspectives

Further investigation of these preliminary observations will have to solve the underlying biological mechanisms and search for concrete evidence in experimental studies. This is mandatory to correctly advise old patients with osteoarthritis on what to eat every day and to integrate clinical practice with valuable prehabilitation interventions. Although increasing the consumption of a certain food may boost health status, chronic eating of one specific food or dietary monotony can in fact be detrimental in older persons [33,34]. Lacking concrete evidence, healthcare professionals must provide cautious nutritional advice, without forgetting that less is more, and that value lies in multimodality.

## Figures and Tables

**Figure 1 nutrients-15-04868-f001:**
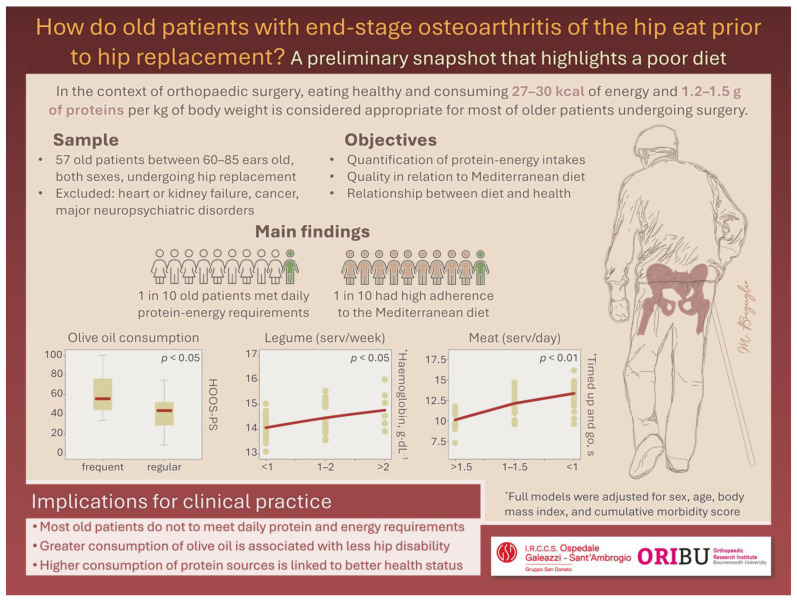
Summary slide of the cross-sectional study investigating diet quantity, diet quality, and their associations with clinical conditions of older patients with end-stage osteoarthritis undergoing elective hip replacement. Diet quantity was assessed through the recall of food and beverages consumed in the last 24 h (24HR) and diet quality was obtained with the Medi-Lite index [12].

**Table 1 nutrients-15-04868-t001:** Characteristics of the study cohort.

Descriptors	Cohort
Sex, male; female	30; 27
Age, years	71.48 (6.54) [60.89–84.14]
BMI, kg·m^−2^	27.00 (4.72) [19.57–40.49]
CCI	3.44 (1.20) [2.00–7.00]
Dietary assessment	
Proteins·kg^−1^	0.75 (0.28) [0.31–1.57]
Kcalories·kg^−1^	19.63 (6.01) [9.79–36.47]
Medi-Lite index	10.37 (2.03) [6.00–15.00]
Laboratory parameters	
RBCs, 10^6^·μL^−1^	4.72 (0.48) [3.48–5.87]
Haematocrit, %	42.78 (3.78) [31.80–52.60]
Haemoglobin, g·dL^−1^	14.30 (1.24) [10.70–17.50]
MCV, fL	90.81 (4.75) [76.60–102.40]
MCH, pg	30.38 (1.77) [24.10–34.10]
MCHC, g·dL^−1^	33.45 (1.04) [30.40–35.60]
Iron, μg·dL^−1^	95.08 (28.41) [45.00–168.00]
Transferrin, mg·dL^−1^	240.86 (26.25) [181.00–307.00]
Ferritin, ng·mL^−1^	150.56 (129.82) [13.00–743.00]
WBCs, 10^3^·μL^−1^	7.21 (1.99) [3.64–12.94]
Lymphocytes, 10^3^·μL^−1^	1.89 (0.62) [0.59–4.18]
Lymphocytes, %	26.86 (7.20) [5.00–40.80]
Transthyretin, mg·dL^−1^	25.35 (4.59) [17.80–37.10]
Albumin, g·dL^−1^	4.42 (0.35) [3.60–5.20]
Total proteins, g·dL^−1^	6.86 (0.48) [5.90–7.80]
C-reactive protein, mg·L^−1^	0.37 (0.51) [0.04–2.76]
Functional status	
Barthel index	98.94 (1.94) [89.00–100.00]
Timed up and go, s	12.45 (3.78) [7.15–23.74]
HOOS-PS	47.36 (19.43) [8.80–100.00]
SF-12 PCS	34.96 (7.58) [24.00–53.60]

Notes: Values are mean (SD) [min–max]. Acronyms: BMI = body mass index; CCI = Charlson comorbidity index; RBCs = red blood cells; MCV = mean corpuscular volume; MCH = mean corpuscular haemoglobin; MCHC = mean corpuscular haemoglobin concentration; WBCs = white blood cells; HOOS-PS = hip disability and osteoarthritis outcome score—physical function; SF-12 PCS = physical component score of the 12-item short form survey.

**Table 2 nutrients-15-04868-t002:** Mediterranean dietary scores and adjusted predictions of laboratory parameters and functional scores.

Dependent Variables	Independent Variables of the Regression Models Adjusted for Sex, Age, BMI, and CCI				
	Legumes Score	Cereals Score	Meat Score ^1^	Dairy Score ^1^	Olive Oil Score
Laboratory parameters					
Haemoglobin, g·dL^−1^	+0.498 *	-	-	-	-
Total proteins, g·dL^−1^	-	−0.356 **	-	+0.276 **	-
Functional scores					
Timed up and go, s	-	-	+1.898 **	-	−2.783 **
HOOS-PS	-	-	+11.352 **	-	−13.618 *

Notes: Values are slope coefficients reported as unstandardised Beta coefficients and statistical significance (* *p* < 0.05, ** *p* < 0.01). Coefficients indicate a change in the dependent variable for a one-unit change in the value of the independent variable. ^1^ Conversely to legumes and cereals, the Medi-Lite tool assigns higher scores to the questions on meat and dairy consumption the lower the consumption. Acronyms: HOOS-PS = hip disability and osteoarthritis outcome score—physical function.

## Data Availability

Data reported in this article are shared, after deidentification, immediately and indefinitely as Appendix A to the publication.

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
