# Peer review of "How Do Older Patients with End-Stage Osteoarthritis of the Hip Eat Prior to Hip Replacement? A Preliminary Snapshot That Highlights a Poor Diet"

_nutrients, 2023, doi:10.3390/nu15234868_

Round 1

Reviewer 1 Report

Comments and Suggestions for Authors

The authors focus on the eating patterns in elderly patients with end-stage osteoarthritis before hip replacement. The study is fascinating, highly needed, and fills the void in the field. The paper is well-designed, and the manuscript is well-written. The discussion is well conducted. References are actual and well chosen. However, the work has shortcomings that should be filled in and some errors to be corrected.

-The descriptions of the tables should be more precise (especially table one; please describe what is in brackets, etc.)

-Always indicate towards which group significance is intended.

- The whole result section should be more transparent – please consider rewriting it

- Please separate the Study Limitation section from the discussion

- References should appear in order. Please check positions 22 and 23.

Comments on the Quality of English Language

Minor editing is required.

Author Response

Dear Reviewer,

Thank you for your commitment to reviewing the manuscript and for the positive and constructive comments.

- We improved the note sections to better explain tables and the figure (line 157, 204, and 320).

- This comment is similar to the one from another Reviewer. We rewrote the results accordingly keeping as suggested by the other Reviewer only p and R^2.

- We separated the limitations from the discussion

- We corrected the formatting error that misplaced the references.

Thank you again for taking the time to read and comment. I renew my wishes that the new corrected version will satisfy the comments and thus make the manuscript worthy of publication.

Best regards
The authors

Reviewer 2 Report

Comments and Suggestions for Authors

It is a preliminary study on the quality and quantity of food received 24 hours before admission for hip arthroplasty as well as on their functional status. The study included 57 only patients, females and males, being between 60 to 84 years old and having a BMI between 19.57 to 40.49.

on page 8 authors used a graphic abstract in an unspecified red? color as Figure 1. It is impossible to read it. Instead they must put in the text some histogram of high quality to clarify theie findings, including those on meat and olive oil consumption

I understood that it is a preliminary study; but 57 patients, with a huge difference in the results [as is the values of BMI] and the consumption of meat or olive oil made their results/conclusions questionable

Do we have any data on the outcome of the operation itsself and the follow-up? were there any postoperative deaths; and how are they related to pre-operative nutritional status?

The aim of the study, as presented in Introduction, L 55-57 is not exaclty the same as that presented at the beginning of Discussion, L. 219-222. The later sounds more clear in relation to the former, which should be the main point.

On L 244-246 authors say: "females reported to ingested approximately 244 1396 kcalories and 55 g of proteins versus 1489 kcalories and 58 g of proteins in males. 245 These intakes are lower than a previous report involving American older adults [10]". Given the variation in body weight, it would be preferred kcals and grams of protein to be express per kilo of body weight.

Discussion should be "cut" in paragraphs. From line 219 up to 288 it is a hole paragraph, making the reader to deny  reading.

Discussion should have comparison of the findings with those of others.

Conclusion should be to the point and not repeat the discussion. Conclusion means presentation [in the form of headings] of MY results and the meaning of MY results for to give "food for thought". Thus, references [that is results of others] have no place in conlusion [L 339].

Author Response

Dear Reviewer,

Thank you for your willingness to review the manuscript.

  • Figure 1: Since there are no limitations on the use of colors in the Figures for this Journal, we preferred using the colors of our institutions. Also, we followed the guidelines for brief research reports of this Journal which limit the number of tables and figures. Therefore, we thought it would be useful to combine multiple graphs into a single figure to make information visible to the reader. We are happy to be able to increase the number of figures and graphs with the approval of the Editorial Office. In the meantime, we made the summary Figure of the manuscript more readable, hoping it can be in line with your comment.
  • There may be differences in the results, BMI (no eligibility criteria for BMI), and other variables, but all analyses and reported findings have been controlled for sex, age, BMI, and cumulative morbidity score (as also detailed in the figure). Therefore, the results are definite from a statistical point of view. We added in the limitations of the small sample size.
  • The nature of the report is specified as a preliminary snapshot of baseline findings. We do not currently have post-operative results to be analyzed in relation to pre-operative results. This concept has been clarified in the study design (line 80).
  • The aim of the study was “to explore the eating habits of old patients undergoing elective hip replacement and examine the existence of an association between diet quantity, quality, and health status.” It was similarly rephrased at the beginning of the discussion “…in order to 1) quantify whether protein and energy intakes were sufficient, 2) verify how much their dietary lifestyle adhered to the Mediterranean diet, and 3) investigate whether there was a relationship between diet and clinical condition.”. Diet quantity was point 1, diet quality was point 2, and health status was point 3. We rephrased the aim of the study in the introduction to make it clearer for readers (line 60).
  • We agree with your comment. The sentence pointed out that we found sexual differences in protein-energy intakes per kilo of body weight. This was reported in the first lines of the results section. We attributed this difference to the diverse body weight in males compared to females. In fact, it seems that absolute intakes were rather similar. As suggested, we added the kcals and grams of proteins expressed per kilo (line 232).
  • We broke the discussion section down into paragraphs and separated the limitations section.
  • Discussion has been integrated with comparisons between our findings and those of others (highlighted in turquoise).
  • The conclusion has been rewritten to keep a brief conclusive message as also suggested by another Reviewer (line 339). A future directions section has been added to give readers some hints for future research (line 344).

Thank you again for taking the time to read out. I renew my wishes that the new corrected version will satisfy the Reviewers and thus make the manuscript worthy of publication.

Best regards
The authors

Reviewer 3 Report

Comments and Suggestions for Authors

The brief report by Bringuglio et al. describes poor diet in older patients with end-stage OA of the hip. The manuscript is well-written and sensible in its structure. Figure 1 is illustrative, although a bit “poster conference-like” in appearance. Below are some specific comments and suggestions for the authors. 

Introduction:

The Introduction is a bit short. The authors could consider expanding it a bit with more background information about the impact of different types of diets on recovery in relation to surgery and OA.

Line 53: The last part of the sentence after the comma does not make sense. Please clarify or rephrase.

M&M:

Line 60: “This article” should be rephrased to “this study”.

Line 72: This statement indicates the results are a part of a larger study. If so, please reference where the “remaining” data are published. If not, please ignore this comment.

Results:

Consider removing other statistical values than the p-value and R-squared to enhance the readability. 

Discussion:

It is appreciated the discussion starts with a build-up to the overall aim of the study. The general structure of the discussion is sensible.

Conclusion:

The conclusion is too long. It seems a bit unfocused and is simply trying to grasp too much. Most of the information can be incorporated into the discussion. Consider limiting the discussion to 3-5 sentences to make it short and concise.

Alternatively, add a paragraph about “future perspectives” to encompass that under its own heading.

Author Response

Dear Reviewer,

Thank you for your willingness to review the manuscript and for the constructive comments.

- Introduction: The introduction was written concisely given the nature of the brief report. However, we agree with the possibility of adding some information. We also clarified the phrase in line 60.

- M&M: line 65: We rephrased. Line 80: We clarified where we are with the processing.

- Results: We enhanced the readability of the results section as suggested.

- Conclusion: The conclusion has been summarized and a future directions section has been added as suggested.

Thank you again for taking the time to read out. I renew my wishes that the new corrected version will satisfy the Reviewers and thus make the manuscript worthy of publication.

Best regards
The authors